

# CO₂ and CH₄ budgets and global warming potential modifications in *Sphagnum*-dominated peat mesocosms invaded by *Molinia caerulea*

Fabien Leroy[1,2,3], Sébastien Gogo[1,2,3], Christophe Guimbaud[4,5], Léonard Bernard-Jannin[1,2,3], Xiaole Yin[6], Guillaume Belot[4,5], Wang Shuguang[6], Fatima Laggoun-Défarge[1,2,3]

[1]Université d'Orléans, ISTO, UMR 7327, 45071, Orléans, France
[2]CNRS, ISTO, UMR 7327, 45071 Orléans, France
[3]BRGM, ISTO, UMR 7327, BP 36009, 45060 Orléans, France
[4]Université d'Orléans, LPC2E, UMR 7328, 45071, Orléans, France
[5]CNRS, LPC2E, UMR 7328, 45071, Orléans, France
[6]School of Environmental Science and Engineering, Shandong University, Jinan, China

*Correspondence to*: Fabien Leroy (fabien.leroy@univ-orleans.fr)

**Abstract.** Plant communities play a key role in regulating greenhouse gas (GHG) emissions in peatland ecosystems and therefore in their ability to act as carbon (C) sinks. However, in response to global change, a shift from *Sphagnum* to vascular plant-dominated peatlands may occur, with a potential alteration in their C-sink function. To investigate how the main GHG fluxes ($CO_2$ and $CH_4$) are affected by a plant community change (shift from dominance of *Sphagnum* mosses to vascular plants, i.e. *Molinia caerulea*), a mesocosm experiment was set up. Gross primary production (GPP), ecosystem respiration (ER) and $CH_4$ emission models were used to estimate the annual C balance and global warming potential under both vegetation covers. While the ER and $CH_4$ emission models estimated an output of, respectively, 376±108 and 7±4 gC m$^{-2}$ y$^{-1}$ in *Sphagnum* mesocosms, this reached 1018±362 and 33±8 gC m$^{-2}$ y$^{-1}$ in mesocosms with *Sphagnum rubellum* and *Molinia caerulea*. Annual modelled GPP was estimated at -414±122 and -1273±482 gC m$^{-2}$ y$^{-1}$ in *Sphagnum* and *Sphagnum + Molinia* plots, respectively, leading to an annual $CO_2$ and $CH_4$ budget of – 30 gC m$^{-2}$ y$^{-1}$ in *Sphagnum* plots and of -223 gC m$^{-2}$ y$^{-1}$ in *Sphagnum + Molinia* ones (i.e., a C-sink). Even if, $CH_4$ emissions accounted for a small part of the gaseous C efflux (ca. 3%), their global warming potential value makes both plant communities have a climate warming effect. The shift of vegetation from *Sphagnum* mosses to *Molinia caerulea* seems beneficial for C sequestration at a gaseous level. However, roots and litters of *Molinia caerulea* could provide substrates for C emissions that were not taken into account in the short measurement period studied here.

## 1 Introduction

Peatlands are wetlands that act as a carbon (C) sink at a global scale. They cover only 3% of the land area but have accumulated between 473 to 621 Gt C (Yu et al., 2010) representing 30% of the global soil C. The C-sink capacity of northern peatlands is closely linked to environmental conditions and plant cover characteristics which limit the activity of soil decomposers. As a result, in spite of the relatively small net ecosystem production in peatlands, the imbalance between primary production and decomposition is enough to allow high organic matter (OM) accumulation as peat (Bragazza et al., 2009). The major





component of peat is the accumulation of *Sphagnum* litter (Turetsky, 2003). *Sphagnum* mosses have a key role in peat accumulation by creating acidic, nutrient poor, wet and anoxic conditions and generating recalcitrant litters. Thus, *Sphagnum* species are able to outcompete vascular plants and reduce microbial decomposition (van Breemen, 1995). However, due to global change, environmental modifications (nutrient input, water table drop, warmer climate, etc.) are expected to cause a

plant community shift in peatlands with an increase in vascular plants (especially graminoids) to the detriment of *Sphagnum* species (Berendse et al., 2001; Buttler et al., 2015; Dieleman et al., 2015). Vascular plant invasion could lead to a faster decomposition of peat OM due to a change in litter quality as a substrate for decomposers, thereby decreasing C-sequestration (Strakova et al., 2011). Furthermore, OM already stored in deep peat may be subject to increased decomposition through the stimulating effect of rhizospheric C input. If these losses are not compensated by an increased gross primary productivity,

peatlands could shift from a sink to a source of C and could increase greenhouse gas emissions, mainly carbon dioxide ($CO_2$) and methane ($CH_4$). Vascular plant invasion in peatlands has mostly been studied through a change in decomposition rates (Moore et al., 2007; Gogo et al., 2016) and modification in decomposer activities (Krab et al., 2013; Strakova et al., 2011). Some studies have paid attention to $CH_4$ emissions with and without the presence of *Carex* or *Eriophorum* (Noyce et al., 2014; Green and Baird, 2012; Greenup et al., 2000) and to $CO_2$ fluxes with different plant community compositions (Neff and

Hooper, 2002; Ward et al., 2013). In spite of observed changes in C fluxes, the role of vascular plant invasion on the C balance in peatlands remains to be elucidated. The aim of this study was to investigate how an invading graminoid species, *Molinia caerulea*, can affect the Greenhouse Gases C Budget (GGCB) of a *Sphagnum*-dominated peatland. *Molinia caerulea* encroachment is well acknowledged problem in Europe linked to anthropogenic pressures such as nutrient deposition and management practices but studies of the effects on peatland ecosystem are still limited (Ritson et al., 2017, Berendse et al.,

2001, Chambers et al., 1999). Here, $CO_2$ fluxes and $CH_4$ emissions were regularly measured in *Sphagnum*-peat mesocosms with and without *Molinia caerulea* during fourteen months and were related to biotic and abiotic factors to estimate the annual C budget.

## 2 Materials and methods

### 2.1 Experimental design, sampling and methods

Twelve cylindrical peat mesocosms (30 cm in thickness and diameter) and water were collected in La Guette peatland (France) in March 2015. The site is a *Sphagnum*-dominated transitional fen that has been invaded by *Molinia caerulea* and *Betula spp* (*Betula verrucosa* and *Betula pubescens*) promoted by hydrological disturbances and nutrient inputs (Gogo et al., 2011). The mesocosms were buried near the laboratory in mineral soil with a waterproof tarpaulin containing peat water surrounding them. Environmental conditions were monitored with a weather station including solar radiation, relative humidity, air and soil

temperature at 5 and 20 cm depth every 15 minutes. The mesocosms were separated into 2 treatment groups: 6 mesocosms containing only *Sphagnum rubellum* (called '*Sphagnum'* plots), and 6 containing both *Sphagnum rubellum* and *Molinia caerulea* (called '*Sphagnum + Molinia'* plots). *Molinia caerulea* appeared in May and increased up to 60% of mesocosms on



average until its senescence in November (Leroy et al., 2017). *Molinia caerulea* seedlings were manually removed from *Sphagnum* plots. The water table level (WTL) was measured by a piezometer installed within each mesocosm and was maintained between 5 and 10 cm depth. The number and height of *Molinia caerulea* leaves were measured.

**2.2 Greenhouse gas measurements**

Measurements were performed with the static chamber method from May 2015 to June 2016. The global principle of this method is to pose a hermetic chamber on the mesocosms in order to monitor the gases concentrations inside this chamber from which gas fluxes between soil-atmosphere can be calculated. Here, $CO_2$ and $CH_4$ fluxes were measured once or twice per week during the growing season (April-October 2015 and April-June 2016) and every two weeks during the winter (November 2015- March 2016). $CO_2$ concentrations were estimated using a GMP343 Vaisala probe inserted into a transparent PVC chamber

(D'Angelo et al., 2016; Leroy et al., 2017). This clear chamber was used to measure the net ecosystem exchange (NEE), the balance between gross primary production (GPP; absorption of $CO_2$ by photosynthesis) and ecosystem respiration (ER, release of $CO_2$ into the atmosphere). ER was measured by placing an opaque cover on the chamber to block photosynthesis. The difference between NEE and ER corresponded to the GPP. The measurements lasted a maximum of 5 min and $CO_2$ concentration was recorded every 5 seconds. The slope of the relationship between $CO_2$ concentration and time allowed fluxes

(in µmol $CO_2$ m$^{-2}$ s$^{-1}$) to be calculated. $CH_4$ emissions were measured using SPIRIT, a portable infrared laser spectrometer (Guimbaud et al., 2016), measuring $CH_4$ concentration in a transparent chamber. Measurements take several to twenty minutes with time resolution of 1.5 s (Guimbaud et al., 2011).

**2.3 Carbon flux modelling**

**2.3.1 Ecosystem Respiration**

The ER increased with increasing air temperature and decreasing WTL in both vegetation covers (Supplementary material), as found by Bortoluzzi et al. (2006). To derive ER for the entire year, the same equation as these authors were used for *Sphagnum* plots (Eq. 1):

$$ER_{sph} = \left[ a * \frac{WTL}{WTL_{ref}} + b \right] * \left( \frac{(T_a - T_{min})}{(T_{ref} - T_{min})} \right)^c \qquad (1)$$

ER is the ecosystem respiration flux (µmol $CO_2$ m$^{-2}$ s$^{-1}$). $T_{ref}$ is the reference air temperature and $T_{min}$ the minimum air

temperature. These two parameters were set as in Bortoluzzi et al. (2006) at 15°C and -5°C, respectively. $T_a$ refers to the measured air temperature (°C). The reference for the WTL (WTL$_{ref}$) was set at -15cm corresponding to the deepest WTL recorded in the mesocosms. The coefficients a, b and c (temperature sensitivity parameters) are empirical parameters.

In *Sphagnum + Molinia* plots, ER was significantly correlated to the number of *Molinia caerulea* leaves (r$^2$=0.44; Supplementary material). Following Bortoluzzi et al. (2006) and Kandel et al. (2013), we included, in addition to WTL and



temperature, a vegetation index based on the number of *Molinia caerulea* leaves in the ER model for *Sphagnum + Molinia* plots (Eq. 2):

$$ER_{mol} = \left[ (a * \frac{WTL}{WTL_{ref}}) + (b * Mc_{leaves}) \right] * \left( \frac{(T_a - T_{min})}{(T_{ref} - T_{min})} \right)^c \tag{2}$$

$Mc_{leaves}$ is the number of *Molinia caerulea* leaves.

### 2.3.2 Gross primary production

The relationship between GPP and photosynthetic photon flux density (PPFD) is often described by a rectangular hyperbola saturation curve with:

$$GPP = \frac{i * PPFD * GPP_{max}}{i * PPFD + GPP_{max}} \tag{3}$$

where i (µmol $CO_2$ µmol $^{-1}$ photon) is the initial slope of the hyperbola, $GPP_{max}$, the maximum GPP (µmol m$^{-2}$ s$^{-1}$) and PPFD, the photosynthetic photon flux density (µmol m$^{-2}$ s$^{-1}$). This approach was modified by Mahadevan et al. (2008) and Kandel et al. (2013) to include the effect of temperature and vegetation on the GPP model. The same equation was used in this study with (Eq. 4):

$$GPP = \frac{GPP_{max} * PPFD}{k + PPFD} * Mc_{leaves} * Tscale \tag{4}$$

where $GPP_{max}$ (µmol m$^{-2}$ s$^{-1}$) represents the GPP at light saturation, the parameter k (µmol m$^{-2}$ s$^{-1}$, Eq. 4) is the half saturation value and $Mc_{leaves}$ is the number of *Molinia caerulea* leaves. $T_{scale}$ is the temperature sensitivity of photosynthesis based on Kandel et al. (2013) and calculated as:

$$T_{scale} = \frac{(T - T_{min})(T - T_{max})}{(T - T_{min})(T - T_{max}) - (T - T_{opt})^2} \tag{5}$$

where $T_{min}$, $T_{opt}$ and $T_{max}$ represent the minimum, optimum and maximum air temperature for photosynthesis and were set at 0, 20 and 40°C, respectively.

### 2.3.3 CH₄ emissions

The CH₄ emissions were significantly correlated to the soil temperature and the water table level (Supplementary material). An equation similar to Eq. 1 was used to model the emissions (Eq. 6):

$$CH_4 = \left[ d * \frac{WTL}{WTL_{ref}} + e \right] * \left( \frac{(T_s - T_{min})}{(T_{ref} - T_{min})} \right)^f \tag{6}$$

where $WTL_{ref}$, $T_{min}$, $T_{ref}$ and $T_{min}$ were set as for the ER equation. $T_s$ refers to the measured soil temperature (°C).

### 2.3.4 Models calibration and validation

Two, randomly select, thirds of the ER and CH₄ emission measurements were used to calibrate the equations and the other third was used for validation in order to verify the calibrated model. Calibration of the GPP models were done using additional





measurements with nets decreasing the irradiance (allowing to have 6 GPP measurements under different luminosity per mesocosms) in order to calibrate the $GPP_{max}$ and k parameters based on the Michaelis-Menten equation. In this ways, all measurement points were used to validate the model. Model quality was evaluated using the determination coefficient ($r^2$) and the Normalized Root Mean Square Error (NRMSE) calculated as:

$$NRMSE = 100 * \frac{\sqrt{(\frac{\sum(y-\hat{y})^2}{n})}}{\bar{y}} \tag{7}$$

where y is the measured value, $\hat{y}$ the computed value, n the number of values and $\bar{y}$ the average of the measured value. The NRMSE indicates the percentage of variance between the measured and the predicted values.

The parameters of ER (a, b and c) and $CH_4$ emissions (d, e and f) models were calibrated by minimizing the NRMSE using the "SANN" method of the optim function in R (R Core Team, 2016).

## 2.3.5 Greenhouse Gases C Budget and global warming potential

The net ecosystem C balance (NECB) represents the net rate of C accumulation or release in or from the ecosystem (Chapin et al., 2006) and is calculated as:

$$NECB= -GPP+ER+F_{CH4}+F_{CO}+F_{VOC}+F_{DIC}+F_{DOC}+F_{PC} \tag{8}$$

where GPP is the gross primary production (μmol $m^{-2}$ $s^{-1}$), ER, the Ecosystem Respiration (μmol $m^{-2}$ $s^{-1}$) and $F_{CH4}$, $F_{CO}$, $F_{VOC}$, $F_{DIC}$, $F_{DOC}$, $F_{PC}$, the fluxes in μmol $m^{-2}$ $s^{-1}$ of methane ($CH_4$), C monoxide (CO), volatile organic C (VOC), dissolved inorganic C (DIC), dissolved organic C (DOC) and particulate C (PC), respectively. In this study, we used a simplified approach based on the GPP, ER and $CH_4$ emissions that we referred as the Greenhouse Gases C Budget (GGCB, gC $m^{-2}$ $y^{-1}$). To calculate annual emissions, we run our models with 15 minutes time step using continuous weather and vegetation data.

The global warming potential over 100 years ($GWP_{100}$; g $CO_2$ eq $m^{-2}$ $y^{-1}$) was calculated for both plant communities based on the annual GHG fluxes (GPP and ER and the $CH_4$ emissions) with the Eq. (9):

$$GWP_{100} = (x + y) * \frac{Molecular\ weight\ of\ CO_2}{Molecular\ weight\ of\ C} + z * \frac{Molecular\ weight\ of\ CH_4}{Molecular\ weight\ of\ C} * GWP_{100}\ of\ CH_4 \tag{9}$$

With x and y representing the annual GPP and ER fluxes (in gC $m^{-2}$ $y^{-1}$), z the annual $CH_4$ emissions (in gC $m^{-2}$ $y^{-1}$). The radiative force ($GWP_{100}$) of $CH_4$ is 34 times that of $CO_2$ (Myhre et al., 2013).

### 2.4 Statistics

The effects of *Molinia caerulea* were assessed by comparing *Sphagnum + Molinia* plots to *Sphagnum* plots with two-ways repeated-measure ANOVAs (with plant cover and date as factors).




## 3 Results

### 3.1 Environmental conditions

The environmental conditions of our measurements did not significantly differ between *Sphagnum + Molinia* and *Sphagnum*

plots (Table 1). The only significant differences concerns the GHG fluxes with more important fluxes in *Sphagnum + Molinia*

5   plots compared to the *Sphagnum* plots.

**Table 1: Mean values of 12 months' measurements of net ecosystem exchange (NEE), gross primary production (GPP), ecosystem respiration (ER), CH₄ emissions (CH₄), photosynthetic active radiation (PAR), water table level (WTL) and air temperature (Ta) in *Sphagnum + Molinia* and *Sphagnum* plots. Significant differences of two-way repeated-measure ANOVAs are expressed as \*p < 0.05, \*\*p < 0.01, \*\*\*p < 0.001 (n = 6). Data are presented as mean ±SE, n =12.**

| | Mean | | Significance |
|---|---|---|---|
| | *Sphagnum* | *Sphagnum + Molinia* | |
| **GHG fluxes** | | | |
| NEE ($\mu$mol m$^{-2}$ s$^{-1}$) | -1.15 ± 0. 25 | -4.63 ± 1.72 | *** |
| GPP ($\mu$mol m$^{-2}$ s$^{-1}$) | -2.25 ± 0. 40 | -7.19 ± 2.28 | *** |
| ER ($\mu$mol m$^{-2}$ s$^{-1}$) | 1.10 ± 0. 37 | 2.56 ± 0. 74 | *** |
| CH$_4$ ($\mu$mol m$^{-2}$ s$^{-1}$) | 0.028 ± 0. 013 | 0.093 ± 0. 005 | *** |
| **Environmental parameters** | | | |
| WTL (cm) | -5.00 ± 0. 70 | -6.81 ± 0.63 | |
| PAR ($\mu$mol m$^{-2}$ s$^{-1}$) | 707 ± 159 | 669 ± 160 | |
| Ta (°C) | 12.27± 2.44 | 12.37 ± 2.49 | |





## 3.2 Measured GHG fluxes

ER was significantly higher in *Sphagnum + Molinia* plots compared to *Sphagnum* ones. In both vegetation covers, the ER was maximum in July and minimum in January-February (Table 1, Fig. 1a). GPP increased during the vegetation period (linked to the number of *Molinia* leaves), whereas in *Sphagnum* plots the GPP was relatively constant (Fig. 1b). After the senescence of

5  *Molinia caerulea*, the GPP did not differ between the two treatments, unlike ER that remained higher in *Molinia* plots compared to *Sphagnum* ones. As a result, the NEE was higher in *Sphagnum + Molinia* plots than in *Sphagnum* ones during the growing season, but was lower the rest of the time (Fig. 1c). $CH_4$ emissions significantly increased in *Sphagnum + Molinia* plots with a peak of emissions in summer (June to August) and the lowest emissions in winter (Fig. 1d).

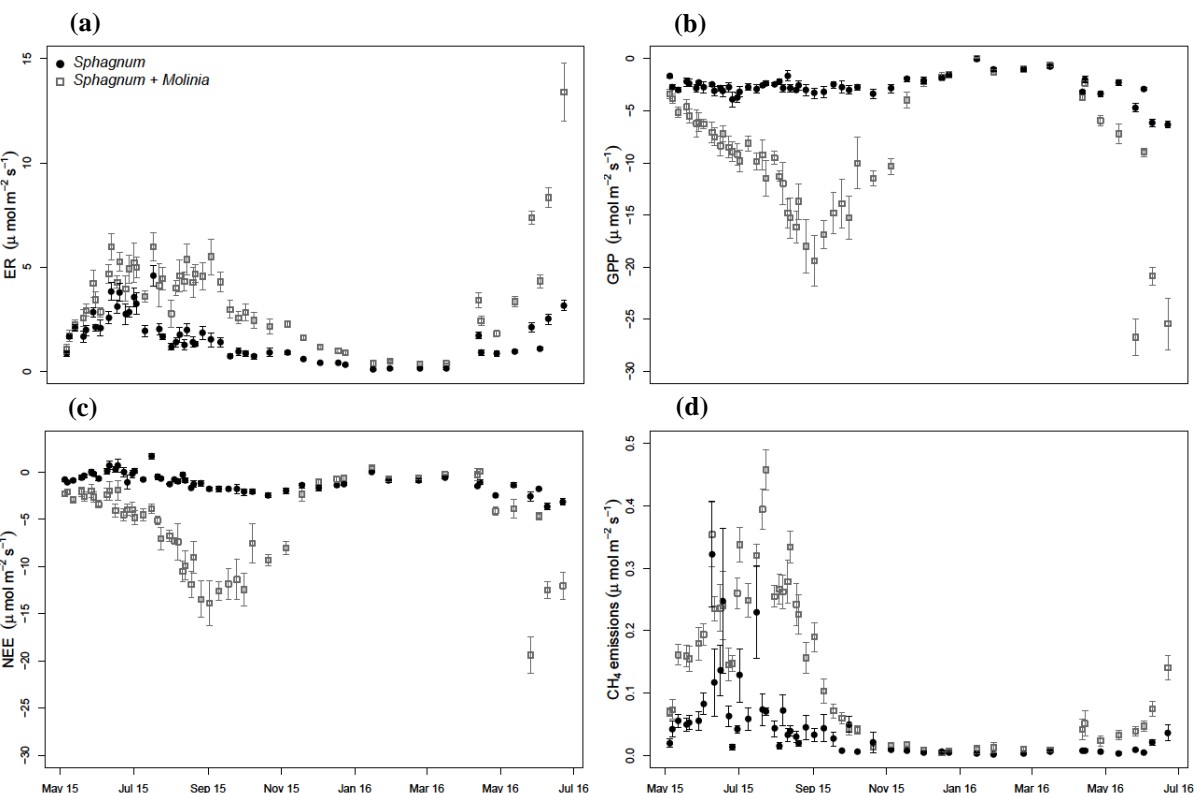

**Figure 1: Measurements of ecosystem respiration (ER; a), gross primary production (GPP, b), net ecosystem exchange**
10  **(NEE, c) and $CH_4$ emissions (d) in *Sphagnum* and *Sphagnum + Molinia* plots (±SE, n=6) from May 2015 to June 2016.**

## 3.3 Calibration and validation of the GPP models

GPP parameters were calibrated using the photosynthesis-irradiance curves based on the Michaelis-Menten equation using four additional measurements (Fig. 2). The $GPP_{max}$ decreased from -4.6 to -7.4 µmol $m^{-2}$ $s^{-1}$ in *Sphagnum* plots and from -7.2 in April to -25.7 µmol $m^{-2}$ $s^{-1}$ at the end of June in *Sphagnum + Molinia* plots.





**Figure 2: Dependence of gross primary production (GPP) on irradiance at four dates. The photosynthesis-irradiance curve shows the maximum rate of photosynthesis (GPP$_{max}$) and the half saturation value (k).**

These increases are linked to *Sphagnum growth* and the number of *Molinia caerulea* leaves, respectively (Supplementary materials). The parameter k ($\mu$mol m$^{-2}$ s$^{-1}$, Eq. 4) is the half saturation value and was set at the mean k value of the four dates with a k equal to 259 $\mu$mol m$^{-2}$ s$^{-1}$ for *Sphagnum* plots and 285 $\mu$mol m$^{-2}$ s$^{-1}$ for *Sphagnum + Molinia* ones.

Models validations were done using all the measurements points and showed a good reproduction of the GPP measurements, even if the relatively constant GPP in *Sphagnum* plots had a NRMSE close to 70.




### 3.4 Calibration and validation of the ER and CH₄ emissions models

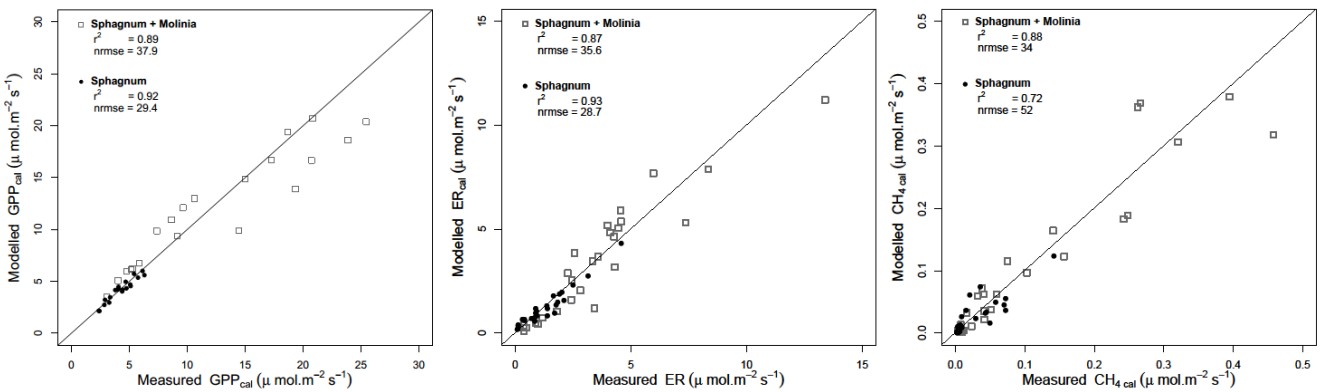

**Figure 3: Calibration of the models by comparison of simulated and measured ecosystem respiration (ER), gross primary production (GPP), and CH₄ emission (CH₄) in *Sphagnum* and *Sphagnum + Molinia* plots. The diagonal lines represent the 1:1.**

Calibration of the models showed a good agreement between the modelled and measured ER and $CH_4$ emissions with a high $r^2$ and low NRMSE for both plant communities (Fig. 3). Regarding the model evaluation, the validation data represented the ER measurements well, especially in *Sphagnum* plots with a $r^2$ of 0.82 and a NMRSE of 46.8 (Table 2). However, in *Sphagnum + Molinia* plots, the ER model validation showed a $r^2$ close to 0.6 but with the higher NMRSE. The validation of the $CH_4$ models explained a good proportion of the variance with a $r^2$ of 0.66 in *Sphagnum* plots and of 0.83 in *Sphagnum + Molinia* plots (Table 2).

| | | Validation | |
|---|---|---|---|
| | | *Sphagnum* | *Sphagnum + Molinia* |
| ER | | | |
| | $r^2$ | 0.82 | 0.59 |
| | NRMSE | 46.8 | 94.7 |
| | a | 2.50 | 1.77 |
| | b | 0.33 | 0.0096 |
| | c | 1.49 | 1.43 |
| GPP | | | |
| | $r^2$ | 0.56 | 0.77 |
| | NRMSE | 69.2 | 50.1 |
| CH₄ | | | |
| | $r^2$ | 0.66 | 0.83 |
| | NRMSE | 78.5 | 41.1 |
| | d | 0.041 | -0.065 |
| | e | 0.001 | 0.092 |
| | f | 3.32 | 5.08 |





**Table 2: r², Normalized Root Mean Square Errors (NRMSE) and adjusted model parameters for calibration of ecosystem respiration (ER), gross primary production (GPP), net ecosystem exchange (NEE) and CH₄ emissions (CH₄) in *Sphagnum + Molinia* and *Sphagnum* plots.**

The model parameters a and c, respectively related to WTL and temperature sensitivity for ER models, were close for both plant communities, ranging for a from 2.50 to 1.77 and for c from 1.49 to 1.43 in *Sphagnum* and *Sphagnum +Molina* plots respectively (Table 2). Concerning the parameters of the CH₄ models, d and f differed between the two treatments. The parameter d connected to WTL was positive at 0.041 in *Sphagnum* plots but negative at -0.065 in *Sphagnum + Molinia* plots. The f value, representing the temperature sensitivity, rose from 3.32 in *Sphagnum* plots to 5.08 in *Sphagnum + Molinia* plots.

**3.5 Greenhouse gases carbon budget and global warming potential**

**Table 3: Modeled annual gross primary production (GPP; gC m⁻² y⁻¹), ecosystem respiration (ER; gC m⁻² y⁻¹) and CH₄ emissions (CH₄; gC m⁻² y⁻¹) in *Sphagnum + Molinia* and *Sphagnum* plots.**

|  | GPP | ER | CH₄ |
|---|---|---|---|
| *Sphagnum* | -414 ± 122 | + 376 ± 108 | + 7 ± 4 |
| *Sphagnum + Molinia* | -1273 ± 482 | + 1018 ± 362 | + 33 ± 8 |

The modeled annual GPP over the studied period represented an input of $414 \pm 122$ gC m⁻² y⁻¹ in *Sphagnum* plots and of $1273 \pm 482$ gC m⁻² y⁻¹ in *Sphagnum + Molinia* plots (Table 3). The ER and CH₄ emissions showed, respectively, an output of $376 \pm 108$ and $7 \pm 4$ gC m⁻² y⁻¹ in *Sphagnum* plots and of $1078 \pm 362$ and $33 \pm 8$ gC m⁻² y⁻¹ in *Sphagnum + Molinia* plots (Table 3).

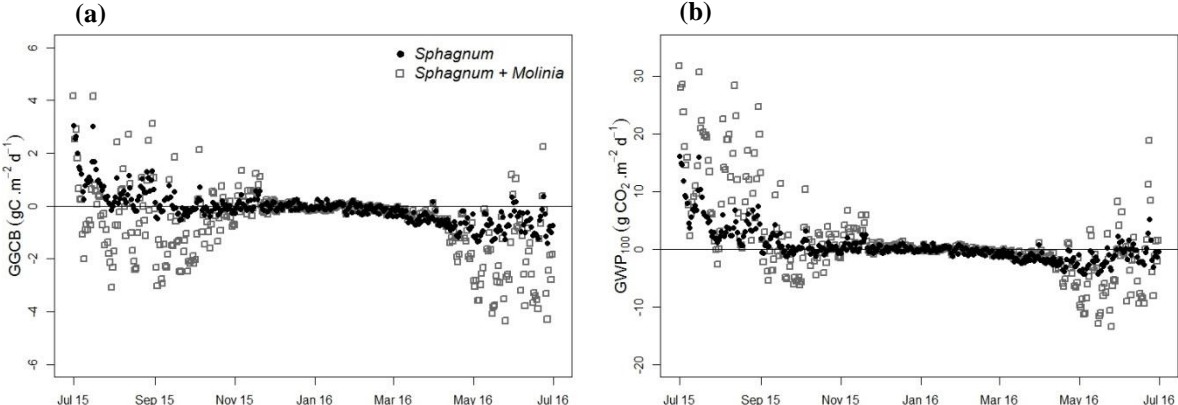

**Figure 4: Greenhouse gases carbon budget (GGCB; a) and global warming potential over 100 years (GWP₁₀₀; b) average per day in *Sphagnum* and *Sphagnum + Molinia* plots.**



From July to December the GGCG was positive in *Sphagnum* plots which means that these plots released more C than they absorbed but the GGCB became negative from January to June (Fig. 4a). In contrast, the GGCB in *Sphagnum + Molinia* plots was mostly negative with positive values only in October and November. It results, the annual GGCB of *Sphagnum* plots absorbed 30 gC m$^{-2}$ y$^{-1}$ whereas the *Sphagnum + Molinia* plots absorbed 223 gC m$^{-2}$ y$^{-1}$. The GWP$_{100}$ for *Sphagnum* and

*Sphagnum + Molinia* plots was, respectively, +195 and +547 g CO$_2$ eq m$^{-2}$ y$^{-1}$.

## 4 Discussion

### 4.1 Gaseous C emissions

The presence of *Molinia caerulea* increased the gaseous C fluxes in the *Sphagnum*-dominated peat mesocosms. Compared to the latter, the GPP was higher with *Molinia caerulea*, with a C uptake close to 1300 gC m$^{-2}$ y$^{-1}$ against 400 gC m$^{-2}$ y$^{-1}$ with

*Sphagnum* alone. The estimated GPP of *Sphagnum* mosses are consistent with studies conducted in boreal peatlands with a GPP close to 350 gC m$^{-2}$ y$^{-1}$ (Peichl et al., 2014; Trudeau et al., 2014). The GPP calculated with *Molinia caerulea* was higher than that measured in the site at La Guette peatland with an average of 1052 gC m$^{-2}$ yr$^{-1}$ (D'Angelo et al., in prep). Such a difference can be explained by the fact that in the field vegetation in collars contained other types of plants such as shrubs and woody chamephytes that exhibited lower GPP (D'Angelo, 2015). A higher GPP of vascular plants is expected to modify the

belowground interactions that are not taken account into our models. Indeed, in comparison to *Sphagnum* mosses, vascular plants have an extensive root system which are able to release C and fuel microbial communities to optimize resource allocation (Fenner et al., 2007). It has been shown that up to 40 % of photosynthates can be allocated to root exudates in peatland (Crow and Wieder, 2005), with half that can be mineralized into CO$_2$ in a week and promote the ER (Kuzyakov et al., 2001) as the root decomposition (Ouyang et al., 2017). The higher ER in mesocosms with *Molinia caerulea* can also be linked to the

metabolism of this vascular plant itself in which leaf respiration can account for more than 40% of the total assimilated C (Kuzyakov et al., 2001). Furthermore, after *Molinia caerulea* senescence, the leaves enhance CO$_2$ emissions through decomposition. Higher CH$_4$ emissions with sedges compared to mosses or shrubs have been explained by the differences in root exudates quality and the aerenchyma of the sedges (e.g. Armstrong et al. 2015).

### 4.2 Models evaluation and sensitivities to parameters

Evaluation showed that our statistical models were efficient in representing ER and GPP for both plant communities. GPP in *Sphagnum* plots was the most difficult variable to represent (Table 2; Fig. 3). It was quite constant in time and only a small decrease was observed in winter when the solar radiation was low. In accordance with Tuittila et al. (2004), the *Sphagnum* growth or cover controlled the photosynthesis. These authors also reported that water saturation of *Sphagnum* govern it photosynthetic capacity and could further improve GPP models (Tuittila et al., 2004). However, with our stable *Sphagnum*

moisture and *Sphagnum* cover, GPP in *Sphagnum* plots was mostly controlled by the PAR. The ER models showed a similar sensitivity in both plant communities to abiotic factors with an empirical factor related to WTL at 2.1 and a temperature





sensitivity close to 1.45 (Table 2). The parameters were similar for both plant communities and ER differences were mainly due to the contribution of *Molinia* leaves to aboveground and belowground respiration (Kandel et al., 2013). Modeling $CH_4$ as $CO_2$ emissions explained a good proportion of the variance (between 70 and 80%). The parameters of the $CH_4$ models differed with vegetation cover. Parameter d connected to the WTL had an opposite sign in the two vegetation covers. This difference

was difficult to interpret as the large variation of parameter e shifted the relationship between parameter d and the WTL. Even so, the presence of *Molinia caerulea* increased the temperature sensitivity of $CH_4$ emissions. Such increase of the temperature sensitivity could result from modification of methanogenesis pathways. Acetoclastic methanogenesis often dominated in minerotrophic peatlands, as La Guette peatland, and required less energy than hydrogenotrophic methanogenesis pathways (Beer and Blodau, 2007).Vascular plants, as *Molinia caerulea*, can influence the methane production through the introduction

of roots exudates in the deep layer by providing substrate availability. Whilst roots exudates are source of acetate and thus suggested to favor acetoclastic methanogenesis (Saarnio et al., 2004), it can also stimulate the decomposition of recalcitrant organic matter favoring hydrogenotrophic methanogenesis (Hornibrook et al., 1997).  Shift from acetoclastic to hydrogenotrophic methanogenesis pathways could explain the increase of the temperature sensitivity observed here. Contributions of methanogens pathways to methane release could be explored by using mechanistic models. Such models

could obtain new insight with additional measurements as substrate supply or microbial community response that could consider in future studies.

### 4.3 Annual C fluxes and GGCB

The shift from *Sphagnum* to *Molinia*-dominated peat mesocosms increased the C fixation through the GPP but also lead to an increase of the annual C output with $CO_2$ and $CH_4$ emissions. The gaseous C balance shows that both plant communities act

as C-sinks with a storage of 30 gC $m^{-2}$ $y^{-1}$ in *Sphagnum* plots and 223 gC $m^{-2}$ $y^{-1}$ in *Sphagnum + Molinia* plots. These results contrast with the assumption mentioned in the introduction, that vascular plants could lead to a decrease in C-sequestration. Nevertheless, the C-sink function of *Molinia*-dominated peat mesocosms can be questioned in view of the biomass production of *Molinia caerulea*. The root production, estimated by Taylor et al. (2001) at 1080 g $m^{-2}$ $y^{-1}$, was produced with current-year photosynthates, meaning that the C-allocation in roots could account for 540 g C $m^{-2}$ $y^{-1}$. Such an amount corresponds to a

larger proportion than the C stored in *Sphagnum + Molinia* plots (223 g C $m^{-2}$ $y^{-1}$) and could represent emission of the C already stored. Furthermore, C stored in roots, litters and leaves of *Molinia caerulea* could contribute to future C emissions by decomposition or respiration not taken into account here. Even with this C-sink function, $GWP_{100}$ is positive for both vegetation covers. Although *Sphagnum + Molinia* plots act more as a C sink than *Sphagnum* ones, the higher $GWP_{100}$ of $CH_4$ compared to $CO_2$ combined with the high emissions of $CH_4$ for *Sphagnum + Molinia* plots lead to a higher contribution of these plots to

the greenhouse effect than in *Sphagnum* ones.

The shift from *Sphagnum* to *Molinia*-dominated peatlands enhanced $CO_2$ uptake by photosynthesis which led to higher $CO_2$ and $CH_4$ emissions. The application of models taking air temperature, water table level and vegetation index into account described these $CO_2$ fluxes and $CH_4$ emissions well. Respiration sensitivity to the two abiotic factors (temperature and WTL)





was similar in both communities. However, the presence of *Molinia caerulea* seems to increase the sensitivity of $CH_4$ emissions to temperature. Modeling the C balance suggested that both *Sphagnum* and *Sphagnum + Molinia* plots acted as a C-sink. However, belowground C allocation as root C stocks needs further consideration due to their potential role as a substantial C source.

**Author contribution.**

FL, SG and FLD designed the experiment.

FL, SG, CG, XY, GB and WS collected data.

FL, SG, CG, LBJ and FLD performed model simulations and data analysis

FL prepared the manuscript with contributions from all co-authors

**Acknowledgements** This work was supported by the Labex VOLTAIRE (ANR-10-LABX-100-01). The authors gratefully acknowledge the financial support provided to the PIVOTS project by the Région Centre – Val de Loire (ARD 2020 program and CPER 2015 -2020). They thank A. Menneguerre for his contribution to gas measurements and P. Jacquet and C. Robert
for their assistance in SPIRIT maintenance. We also thank E. Rowley-Jolivet for revision of the English version.

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
