# Peer review of "CO2 and CH4 budgets and global warming potential modifications in *Sphagnum*-dominated peat mesocosms invaded by *Molinia caerulea"

_Biogeosciences, 2019_

## Short Comment (SC1) · 25 Mar 2019

This manuscript describes a one year mesocosm experiment with two types of vegetation communities sphganum mosses and sphagnum + molinia grasses. CO2 and CH4 fluxes have been measured extensively over the one year period and this MS discusses the vegetation community caused differences in the annual GPP, ER and CH4 fluxes. The topic is important but I have two main concern: 1) the authors don't report if the molinia impacts the sphagnum mosses in any way during the experiment. If the impact of molinia is simply additive it may not describe the "field impact" of dense vascular cover on sphagnum dominated ecosystem and its functions. 2) only 1/3 of the

data used here is "new" and this is not clearly told.

detailed comments: l23: to have

pg1 l27: maybe C storage would be better term than C sink

pg2 l1: this sentence doesn't read well. consider changing into something like: Accumulating Sphagnum litter forms a major component of peat (Turetsky, 2003) and creates acidic... pg2 l9: reference needed for stimulation

pg2 l10, change order of sentences: the increase in greenhouse gas emissions, mainly carbon dioxide (CO2)and methane (CH4) could shift the peatland from a sink to a source of carbon

pg3 l1: how was Molinia removed? with roots and stems?

pg3 l3: how was the wtl controlled?

pg 3 l3: what was the density of the sphagnum mosses in the treatments? did molinia impact sphagnum in any way? Considering the impact on invasion, it is a very different situation if molinia is just added on top of the sphagnum or if the invading molinia affects the sphagnum by decreasing its cover. the studies you refer in introduction imply that vascular plants replace sphganum mosses: A dense vascular plant cover should make the moss layer less dense and therein impact its functions. I am not sure how your treatment is in line with this as I expect that the pure-sphagnum mesocosms have grown under molinia during previous growing seasons. or did you choose such plots that had very little molinia originally? I think this is a really major issue and you should properly explain how this operates in your experiment. so does the moss layer in the two treatments differ in any means? pg3, l10: are you using here the same data that has been published already by Leroy et al 2017, with only GPP added? This feels strange to me. only 1:3 of the data is new. At least you should be clearly stating this.

pg3 l15: Did you measure PAR during your NEE measurements and did the irradiation stay stable during the measurement? using the PAr measured every 15 min might be

fine during clear days, but often it varies quite a lot.

pg3 l20: Explain here that you are improving the data analysis from your previous paper so that you can evaluate the annual flux

2.3.2: Now this is very much unclear that you have actually not used the data collected as explained earlier to calibrate this model, but you use different data that is explained only later. I think you should move the GPP model calibration explanations here and explain the measurement methods much better

pg4 l9: did you measure PPFD?

pg4, l18: what is the measured T? soil? air inside chamber? and how was it measured?

pg4 l21: maybe you should refer here to your previous paper?

pg4, l27: randomly selected

pg5 l1: see previous comment on GPP model

pg5 l16: why did you decide to leave DOC out from this MS though you have measured it?

2.4: maybe open this a bit more. Effect of Molinia to what?

pg6 l4: why does PAR differ between the treatments? did you have a sensor in each mesocosm and at which height? from methods I understood that you have a weather station measuring par, temp, wt variables, but these results make it look like they have been measured from each mesocosm

pg6 l2: why are ghg fluxes mentioned here? replace more important with "higher"

pg7 l2: you already showed this with your previous paper

pg7 l3: delete table 1

pg7 l4: the fig 1b gives impression that sphagnum has not yet stabilized to conditions

without molinia during the first summer, while during the start of next summer GPP is clearly higher. maybe the cover and density of sphagnum has increased as they have grown for a year without any shading molinia? Similarly, the difference in the rate of ER is much more pronounced during the second year, implying that molinia roots in pure-sphagnum plots have decomposed.

pg7 l13: increased (not decreased)

Figure 4 b) Do you think it reasonable to present daily GWP? how did you even obtain the values as in methods you say you calculated them based on annual, not daily flux estimates?

pg11 l1: GGCB

pg11 l1: is it possible that there remained some root of clipped molinia in sphagnum columns that were decaying during the first half of the year?

pg11 l8-10: maybe you want to give some reason, with references why molinia increased GPP. more photosynthesizing plant material, potentially higher photosynthetic capacity? molinia did not decrease sphagnum cover compared to pure sphagnum (true or not?).

pg11 l14: lower GPP or actually lower photosynthetic capacity and lower leaf area?

pg11 l22: Molinia is not a sedge

pg11: 4.2. I am not convinced how interesting is the discussion about the parameter sensitivities, especially for empirical parameters that do not have a clear ecologically meaning full explanation (such as Pmax or half saturation constant), My advice is to shorten this section and leave only meaningful explanations

pg11 l28: how did you validate this?

Pg 11 l30: throughout the MS you use terms irradiation, PPFD and PAR, please choose one of these and use only it

pg12 l2-3: delete "as CO2 emissions", it is bit confusing there

pg12 l4-5: this is one example of unvague text that I would delete: Parameter d connected to the WTL had an opposite sign in the two vegetation covers. This difference was difficult to interpret as the large variation of parameter e shifted the relationship between parameter 5 d and the WTL.

pg 12 l9-10: rephrase: Vascular plants, as Molinia caerulea, can influence the methane production through the introduction of root exudates into the deep peat layer by increasing substrate availability. Also, add a reference for this

pg12 l13-14: based on the above explanations I don't quite buy this. why would it switch to hydrogenotrophic as acetates are provided for acetoclastics?

pg12 l20-21: you could add the references also to here

pg 12 l26-27: but you do have last year's roots there?

pg13 l31-pg14 l4: I don't think you need to repeat this information here but maybe you can give some implications that your study likely has.

―――――――――――――――――――――

---

## Referee Comment (RC1) · Anonymous Referee #1 · 19 Apr 2019

The manuscript presents results of mesocosm study of two sets of vegetation samples, representing two stages of fen development: Sphagnum-dominated community and the one invaded by Molinia caerulea. The empirical models predicting gross primary production, respiration and methane emission are calibrated. The larger productivity and greenhouse gas emissions from Molinia are demonstrated. Despite well-known limitations of using mesocosm-derived vegetation characteristics for natural ecosystems, the study provides useful contribution to our knowledge of carbon budget of wetlands.

I have no general concerns on the paper. There are some specific comments, that hopefully can serve to improvement of the paper quality:

[Figure]

1) I recommend to add a photo of mesocosm experiment setup.

2) *"Molinia caerulea appeared in May and increased up to 60% of mesocosms . . . "*
What is the variable with the value 60%? Area, mass?

3) *"Here, $CO_2$ and $CH_4$ fluxes were measured once or twice per week during the growing season (April-October 2015 and April-June 2016) and every two weeks during the winter (November 2015-March 2016)".* Please speculate on the possible effects of diurnal cycle on long-term averages of carbon budget of samples, which you are missing with this measurement frequency.

4) In eq. (2), I guess, ER should go to zero when Mcleaves=0, as respiration is hardly possible without leaves.

5) *"The only significant differences concerns the GHG fluxes with more important fluxes in Sphagnum + Molinia ..."* Not clear what do you mean by "important" here.

6) *"To calculate annual emissions, we run our models with 15 minutes time step using continuous weather and vegetation data."* Please justify the application of models (1-9) calibrated on daytime measurements only (or may be not only daytime, but you don't indicate the times of measurements in 2.1-2.2 sections) to the annual period.

7) In eq. (6), methane emission is dependent on temperature as $T^f$, whereas in numerous wetland models temperature effect on emission (production) is represented by $q_{10}^{(T/10)}$ term. Please, justify your choice.

8) In Table 1, there are no cases denoted by "*" and "**".

9) In eq. (1) and (2) I would denote a, b and c differently, as they get different values.

10) *"In both vegetation covers, the ER was maximum in July and minimum in January-February (Table 1, Fig. 1a)."* Table 1 does not provide information on seasonality.

11) *"These increases are linked to Sphagnum growth and the number of Molinia caerulea leaves, respectively."* Why GPPmax should depend on leaves area, whereas

the latter is already included in (4) as separate multiplier?

12) *"Parameter d connected to the WTL had an opposite sign in the two vegetation covers. This difference was difficult to interpret as the large variation of parameter e shifted the relationship between parameter d and the WTL."* Please be more elaborate in this explanation, as it is not readily understandable at the moment.

————————————————————

---

## Referee Comment (RC3) · Anonymous Referee #3 · 5 May 2019

**CO$_2$ and CH$_4$ budgets and global warming potential modifications in *Sphagnum*-dominated peat mesocosms invaded by *Molinia caerulea**

by Fabien Leroy *et al*.

Generally, the topic of the manuscript is of interest for readers of Biogeosciences.

Unfortunately, I think the paper should be rejected for the following reasons:

- The methane (CH$_4$) and ecosystem respiration (ER) data has already been published here: Leroy *et al*. (2017): Vegetation composition controls temperature sensitivity of CO$_2$ and CH$_4$ emissions and DOC concentration in peatlands, Soil Biology and Biochemistry 107, 164-167, https://doi.org/10.1016/j.soilbio.2017.01.005
- The general approach and the results of the annual model for ER and gross primary production (GPP) has recently been published in this paper: Leroy *et al*. (2019): Response of C and N cycles to N fertilization in *Sphagnum* and *Molinia*-dominated peat mesocosms, Journal of Environmental Sciences 77, 264-272, https://doi.org/10.1016/j.jes.2018.08.003

Therefore, I do not see any reasons why this manuscript should additionally be published, especially as the authors fail to mention that the data has already been published elsewhere. Most aspects of the discussion are of course similar (change of methanogenic pathways, temperature sensitivity).

The only aspect which has been additionally done is discussing the approach to derive the annual values in more detail and giving actual numbers of annual CH$_4$ fluxes. The approach to model annual balances of CH$_4$ has, however, already been described in the second paper. Although the results are not given there, this does in my opinion not justify an additional publication as the important aspects (mean values, temperature dependency, and correlation with other variables) are already published.

Further, the approach of using temperature (and water table or biomass proxies) for ER and photon flux density for GPP is fairly common when working with manual chamber. Therefore, there are no new aspects compared to the papers quoted by the authors (e.g. Kandel *et al*. 2013) and others using similar approaches which would justify publication.

---

## Author Comment (AC1) · 31 May 2019

The response of the referee 1 comments are enclosed in the supplement part

Please also note the supplement to this comment:
https://www.biogeosciences-discuss.net/bg-2019-82/bg-2019-82-AC1-supplement.pdf

---

## Author Comment (AC2) · 31 May 2019

This manuscript describes a one year mesocosm experiment with two types of vegetation communities sphagnum mosses and sphagnum + molinia grasses. $CO_2$ and $CH_4$ fluxes have been measured extensively over the one year period and this MS discusses the vegetation community caused differences in the annual GPP, ER and $CH_4$ fluxes. The topic is important but I have two main concern: 1) the authors don't report if the molinia impacts the sphagnum mosses in any way during the experiment. If the impact of molinia is simply additive it may not describe the "field impact" of dense vascular cover on sphagnum dominated ecosystem and its functions. 2) only 1/3 of the data used here is "new" and this is not clearly told.

**Detailed comments:**

l23: to have pg1 l27: maybe C storage would be better term than C sink

➢ Yes, it have been modified

pg2 l1: this sentence doesn't read well. consider changing into something like: Accumulating Sphagnum litter forms a major component of peat (Turetsky, 2003) and creates acidic...

➢ It have been modified.

pg2 l9: reference needed for stimulation

➢ Indeed, we added the reference : Girkin et al., 2018

pg2l10,change order of sentences: the increase in greenhouse gas emissions,mainly carbon dioxide (CO2)and methane (CH4) could shift the peatland from a sink to a source of carbon
➢ Done

pg3 l1: how was Molinia removed? with roots and stems?
➢ The young Molinia was carefully removed with roots and stems (added in the text)

pg3 l3: how was the wtl controlled?
➢ Information have been added with : "with addition of peat water when necessary"

pg 3 l3: what was the density of the sphagnum mosses in the treatments?did molinia impact sphagnum in any way? Considering the impact on invasion, it is a very different situation if molinia is just added on top of the sphagnum or if the invading molinia affects the sphagnum by decreasing its cover. the studies you refer in introduction imply that vascular plants replace sphganum mosses: A dense vascular plant cover should make the moss layer less dense and therein impact its functions. I am not sure how your treatment is in line with this as I expect that the pure-sphagnum mesocosms have grown under molinia during previous growing seasons. or did you choose such plots that had very little molinia originally? I think this is a really major issue and you should properly explain how this operates in your experiment. so does the moss layer in the two treatments differ in any means?

➢ Information have been added : "All mesocosms were choose because they was entirely covers by *Sphagnum rubellum* and without *Molinia* stems. *Molinia caerulea* appeared in May and increased up to 60% of mesocosms on average until its senescence in November (Leroy et al., 2017) and did not affect *Sphagnum* cover (unpublished data)."

pg3, l10: are you using here the same data that has been published already by Leroy et al 2017, with only GPP added? This feels strange to me. only 1:3 of the data is new. At least you should be clearly stating this.

> The ER and CH4 emissions data in Leroy et al 2017 are used to compare the temperature sensitivity between the plants communities. Here, we used in different purpose. It is to establish a C balance between the treatments and that why the GPP have been added. A sentence have been added to explant this.

pg3 l15: Did you measure PAR during your NEE measurements and did the irradiation stay stable during the measurement? using the PAr measured every 15 min might be fine during clear days, but often it varies quite a lot.

> The PAR was measured at the beginning and the end of each of the NEE measurement and remain stable during of the measurement.

pg3l20: Explain here that you are improving the data analysis from your previous paper so that you can evaluate the annual flux

> Done

2.3.2: Now this is very much unclear that you have actually not used the data collected as explained earlier to calibrate this model, but you use different data that is explained only later. I think you should move the GPP model calibration explanations here and explain the measurement methods much better

> Information have been added to clarify this point.

pg4 l9: did you measure PPFD?

> The PAR was measured at the beginning and the end of each of the NEE measurement

pg4,l18: what is the measured T ?soil? air inside chamber? And how was it measured?

> T the air temperature measured with the weather station. The information have been added.

pg4 l21: maybe you should refer here to your previous paper?

> Done

pg4, l27: randomly selected pg5 l1: see previous comment on GPP model

> It have been modified

pg5l16: why did you decide to leave DOC out from this MS though you have measured it? 2.4: maybe open this a bit more. Effect of Molinia to what?

> The DOC is not discussed in this MS because we measured its concentration but not its export (due the experiment setup). In this way, the discussion about the difference of DOC would have been close to our previous article.

pg6 l4: why does PAR differ between the treatments? did you have a sensor in each mesocosm and at which height? from methods I understood that you have a weather station measuring par, temp, wt variables, but these results make it look like they have been measured from each mesocosm

➤ The environmental parameters reported in the Table I are those measured during the 15 min of CO2 measurements and measured for each mesocosm.

pg6 l2: why are ghg fluxes mentioned here? replace more important with "higher"

➤ It have been replaced

pg7 l2: you already showed this with your previous paper

➤ Indeed, the ER and the CH4 emissions was showed in the previous paper. However, here we added the GPP and NEE and mostly want to showed these results without need for lecturer to go to our previous paper.

pg7 l3: delete table 1

➤ Done

pg7 l4: the fig 1b gives impression that sphagnum has not yet stabilized to conditions without molinia during the first summer, while during the start of next summer GPP is clearly higher. maybe the cover and density of sphagnum has increased as they have grown for a year without any shading molinia? Similarly, the difference in the rate of ER is much more pronounced during the second year, implying that molinia roots in pure-sphagnum plots have decomposed.

➤ Indeed, there are an increase of fluxes the second year. However, this effect was noticed in both vegetation treatments (not only in pure-sphagnum plots). It can be related to a growth of Sphagnum mosses but also to different environmental conditions during the measurements.

pg7 l13: increased (not decreased)

➤ It decreased from -4.6 to -7.4 µmol m$^{-2}$ s$^{-1}$.

Figure 4 b) Do you think it reasonable to present daily GWP? how did you even obtain the values as in methods you say you calculated them based on annual, not daily flux estimates?

➤ The Fig. 4b was removed. It was not referred in the text

pg11 l1: GGCB

➤ Modified

pg11 l1: is it possible that there remained some root of clipped molinia in sphagnum columns that were decaying during the first half of the year?

➤ When the mesocosms was collected, they only contains only Sphagnum. However, they was also all surrounding by Molinia caerulea, so it is possible that some roots was decaying in plots.

 pg11 l8-10: maybe you want to give some reason, with references why molinia increased GPP. more photosynthesizing plant material, potentially higher photosynthetic capacity? Molinia did not decrease sphagnum cover compared to pure sphagnum (true or not?).

➤ Yes, indeed. This points have developed in the discussion part.

pg11 l14: lower GPP or actually lower photosynthetic capacity and lower leaf area?

> We regrouped both photosynthetic capacity and lower leaf area into the term of GPP because there are lower leaf area of shrubs compared to Molinia caerulea. However, the comparison of their photosynthetic activity have not been performed.

pg11 l22: Molinia is not a sedge

> Right, it have been modified

pg11: 4.2. I am not convinced how interesting is the discussion about the parameter sensitivities, especially for empirical parameters that do not have a clear ecologically meaning full explanation (such as Pmax or half saturation constant), My advice is to shorten this section and leave only meaningful explanations

> With the comment below, this sections was modified

pg11 l28: how did you validate this?

> With the supplementary material Fig. S1a.

Pg11l30: through out the MS you use terms irradiation ,PPFD and PAR, please choose one of these and use only it

pg12 l2-3: delete "as CO2 emissions", it is bit confusing there

> Done

pg12 l4-5: this is one example of unvague text that I would delete: Parameter d connected to the WTL had an opposite sign in the two vegetation covers. This difference was difficult to interpret as the large variation of parameter e shifted the relationship between parameter 5 d and the WTL.

> This part have been removed

pg12l9-10: rephrase: Vascular plants as Molinia caerulea, can influence the methane production through the introduction of root exudates into the deep peat layer by increasing substrate availability.

> Done

Also, add a reference for this pg12 l13-14: based on the above explanations I don't quite buy this. why would it switch to hydrogenotrophic as acetates are provided for acetoclastics? pg12

> The sentences was modified as follows : Whilst roots exudates are source of acetate and thus suggested to favor acetoclastic methanogenesis (Saarnio et al., 2004), the roots exudates also stimulate the decomposition of recalcitrant organic matter favoring hydrogenotrophic methanogenesis (Hornibrook et al., 1997), and maybe more than acetates promoting acetoclastic methanogenesis.

l20-21: you could add the references also to here

> Done

pg 12 l26-27: but you do have last year's roots there?

> At the beginning all mesocosm was only pure-sphagnum plot. During the collection, only few roots was apparent, especially in comparison to the the end of the experiement

where, Molinia+Sphagnum mesocosm contains high roots biomass ( at visual scale, not measurement was performed)

pg13 l31-pg14 l4: I don't think you need to repeat this information here but maybe you can give some implications that your study likely has.

> This paragraphs was here to sum up the informations discussed previously but sime ipmications have been added

---

## Author Comment (AC3) · 31 May 2019

Generally, the topic of the manuscript is of interest for readers of Biogeosciences.

Unfortunately, I think the paper should be rejected for the following reasons:

- The methane (CH4) and ecosystem respiration (ER) data has already been published here: Leroy et al. (2017): Vegetation composition controls temperature sensitivity of CO2 and CH4 emissions and DOC concentration in peatlands, Soil Biology and Biochemistry 107, 164-167, https://doi.org/10.1016/j.soilbio.2017.01.005
- The general approach and the results of the annual model for ER and gross primary production (GPP) has recently been published in this paper: Leroy et al. (2019): Response of C and N cycles to N fertilization in Sphagnum and Molinia-dominated peat mesocosms, Journal of Environmental Sciences 77, 264-272, https://doi.org/10.1016/j.jes.2018.08.003

Therefore, I do not see any reasons why this manuscript should additionally be published, especially as the authors fail to mention that the data has already been published elsewhere. Most aspects of the discussion are of course similar (change of methanogenic pathways, temperature sensitivity).

The only aspect which has been additionally done is discussing the approach to derive the annual values in more detail and giving actual numbers of annual CH4 fluxes. The approach to model annual balances of CH4 has, however, already been described in the second paper. Although the results are not given there, this does in my opinion not justify an additional publication as the important aspects (mean values, temperature dependency, and correlation with other variables) are already published.

Further, the approach of using temperature (and water table or biomass proxies) for ER and photon flux density for GPP is fairly common when working with manual chamber. Therefore, there are no new aspects compared to the papers quoted by the authors (e.g. Kandel et al. 2013) and others using similar approaches which would justify publication.

  > This comments have also been relieved by the referee #2. Sentences have been added in the manuscript to explain the different approach of this manuscript compared to the others papers. Here, our goal was to modelize the C fluxes under different plants communities. The works of modelization have not been developed in Leroy et al., 2017 (which concern only the temperature sensitivities) and in Leroy et al., 2019, which mentioned the number reported here without any support of these numbers.

---

## Author Response (AR1)

**Editor comment**

However, as pointed out by referees #2 and #3, you should explain the difference (what's new and original) in a clearer manner; this is the most important point in examining the revised manuscript.

**Author's Response**

L21: The experimental design and a part of the data have been used in Leroy et 2017 and 2019 to explore the temperature sensitivity and N deposition effect on C and N cycle with two different plants communities in peatlands. In this paper, in addition to GPP data, the novelty was to modelize the C fluxes (GPP, ER and CH4 emissions) and estimated 
[revised manuscript text omitted]

---

## Referee Report (RR1)

Manuscript BG-2019-82

CO2 and CH4 budgets and global warming potential modifications in *Sphagnum*-dominated peat mesocosms invaded by *Molinia caerulea*

Fabien Leroy, Sébastien Gogo, Christophe Guimbaud, Léonard Bernard-Jannin, Xiaole Yin, Guillaume Belot, Wang Shuguang, Fatima Laggoun-Défarge

Leroy and co-authors present a re-analysis of previously published greenhouse gas flux data to model and estimate greenhouse gas budget (for CO2 and CH4) of mesocosms derived from a Spghagnum bog in France subject to invasion by a vascular plant in the genus *Molinia*. The experimental design and gas flux data are shared with two previous publications with the same first author and many of the co-authors. The novelty presented in this study is the modelling of temperature and water and the addition of the vascular plant to produce estimates of net greenhouse gas contributions. Both Sphagnum-only and Sphagnum + Molinia mesocosms were net sources of greenhouse gases, largely due to large CH4 emissions and the large CO2-equivalency of CH4.

Because of the shared underlying dataset, great care needs to be taken in describing exactly how this paper differs from the two previous publications. The question is: does this paper apply something really new to a now twice-investigated year-long series of observations of greenhouse gas flux in mesocosms?

Introduction

The final part of the Introduction makes it clear where the data and analyses of this paper originated and how it differs from previous publications based (at least in part) on the same data set and experiments. However, a GHG budget is potentially a simple calculation based on values of gas flux already reported in the previous publications, particularly Leroy et al. (2017). It could also be a more sophisticated calculation that takes other parameters into account, which might make the publication of those estimates in a separate, stand-alone paper worthwhile. This paper, as stated in the Introduction, appears to take a rather sophisticated approach to estimating GHG budgets from mesocosm data.

I could not find the reference (Leroy et al. 2019) in the References, a critical omission when that paper apparently shares a considerable amount of raw data with the current study. I was able to find a likely candidate through Web of Science. Please include the full citation in the References, and confirm that the paper from Journal of Environmental Sciences is the intended reference.

Materials and methods

Throughout this manuscript, the full name of the studied vascular plant species always given, Molinia caerulea. Because no other species in this genus are discussed, the use of the genus name alone would be suitable for every use after the first, with a comment to the effect of "referred to as Molinia for the rest of this study". This would make the paper a bit easier to read.

The sentence on page 3, lines 18-20 is incomprehensible. Please re-write.
Please quantify what is meant by "several to twenty minutes" on page 3, lines 29-30.

Why were chamber measurements so variable in their time closed? Most users of chamber-based methods standardize the length of time a chamber is closed for measurement. Were the chambers removed when a pre-determined gas concentration value was recorded?

The model based on Bortoluzzi et al. (2016) and Kandel et al. (2013) presented on page 4 (Eq 2) is exactly the same as Eq 2 of Leroy et al. (2019). Table 1 of Leroy et al. (2019) contains many of the same parameters as appear in Table 1 of the current manuscript, yet the values are slightly different. A very similar Table 1 also appears in Leroy et al. (2017), yet again the exact values are different.

Please explicitly describe how these three separate papers using a common (or mostly common?) data set and a single experimental design (slightly modified?) arrived at different values for the components of greenhouse gas fluxes. If the values from previous publications are used in the GGCB calculation of the current manuscript, do the conclusions change at all? Which values are the most accurate - if you were to conduct a meta-analysis of the GHG budget effects of vascular plant invasion of Sphagnum peatlands, which of these three sets of values would you include?

In the description of how the greenhouse gas budget is calculated (Eq. 8), please specify what positive and negative values indicate. Some studies declare negative values to indicate net movement of C into the ground or terrestrial ecosystem (e.g. strong photosynthesis, a net sink), others declare negative values to indicate net movement of C into the atmosphere (e.g. strong respiration, a net source).

Results

Pg 7, LN 3: what do you mean by "linked"? Was GPP significantly correlated with the number of Molinia leaves? Was number of Molinia leaves an important parameter in your GPP model? How was this tested?

Figure 1 has a confusing use of parentheses in the legend. Please change from "(ER; a)" to "(ER); a)" or a similar separation of clauses.
In Figure 1d, it is clear that CH4 emissions from Molinia + Sphagnum plots peaked in summer, as described (Pg 7, LN 7-8). It also appears the Sphagnum plots also had a CH4 emission peak at around the same time. Were the changes in CH4 emissions through the seasons not significant differences in the Sphagnum plots?

Figure 4 appears before it is mentioned in the text. Table 3 appears directly above the block of text mentioning it. Figure and Table position is largely up to the final typesetting, but it was distracting to have these parts of the manuscript appear out of order.

Discussion

Molinia caerulea have aerenchymous tissues, as described by Lloyd et al. (1998). This could be a useful citation to include in the first paragraph of the Discussion on page 11.

Pg 11, LN 26: what is meant by "efficient" here? Did you compare your models to a standard and calculate the proportion of times your models matched the standard?

On page 11, CH4 emissions via aerenchyma is described. On page 12, CH4 emissions via stimulation of methogenic organisms by root exudates from vascular plants is described. On page 13, the increased temperature sensitivity of CH4 emissions under Sphagnum + Molinia is described. Your results show increased CH4 emissions associated with Molinia.
How do these factors fit together?
Kao-Knifflin et al. (2010) appears in the Reference list but nowhere in the text. I suspect it would be a useful place to start in considering how various factors interact to increase CH4 emissions when graminoid plants are added to a Sphagnum bog.

The value of 34 is used here for the GWP100 of CH4. This is apparently from the 2013 IPCC report. Other values of CH4's GWP have been used in various studies, some as low as 25. Do your conclusions - that these wetlands are net contributors to climate warming, and not C-sinks - change with different values of CH4 GWP?

The final sentence on page 13, lines 7-8 is almost word-for-word identical to the final sentence of the immediately preceding paragraph, same page, lines 3-4.
Please write a concluding paragraph that is more than a series of copy-paste from previous parts of the Discussion.

Typos and other minor writing points

Overall, there are more than a few problems with plurals, with missing or added 's' at the ends of many words. This is a most often trivial problem that does not seriously distract from the manuscript and can be fixed with rigorous proof-reading. Careful attention to verb tense is also necessary. The final sentence of the Introduction, on page 2 lines 26 to 28, is a useful example.
"Such C budget calculation allowed the estimation of the global warming potential, a key feature of the paper submitted to Biogeosciences, which was not studied in the previous papers and deserve a communication on its own."
change to
"Such a C-budget calculation has allowed the estimation of the global warming potential, a key feature of the current paper, which has not been studied in the previous papers (i.e. Leroy et al., 2017; 2019) and deserves specific consideration."

Similarly, the sentence on page 4, lines 4-5 can be corrected by swapping the verb tenses of "established" and "derive" - to "and establish a" and "was derived for"

Pg 2, LN 1: conditions should be plural
Pg 2, LN 17: "encroachment is a well known"
Pg 2, LN 19: "ecosystems"
Pg 2, LN 22: change "Leroy et al. 2017 and Leroy et al. 2019" to "Leroy et al. (2017; 2019)"
Pg 2, LN 24: change "in the paper submitted to Biogeosciences" to "in the current paper" and similarly in the sentence on LN 27
Pg 3, LN 6: "chosen"; "were"; "covered"
Pg 3, LN 7-10: repeated sentence
Pg 6, LN 7: the usual term for comparing different greenhouse gases to the effects of carbon dioxide is "global warming potential", or GWP, and usually with a subscript or comment describing the time range (in years) for the comparison.
Pg 11, LN 16: re-arrange some words: "that are not taken into account in our models"

Liturature cited

Kao-Kniffin, J., Freyre, D.S., Balser, T.C., 2010. Methane dynamics across wetland plant species. Aquatic Botany 93, 107–113. doi:10.1016/j.aquabot.2010.03.009

Leroy, F., Gogo, S., Guimbaud, C., Francez, A.J., Zocatelli, R., Défarge, C., Bernard-Jannin, L., Hu, Z., Laggoun-Défarge, F., 2019. Response of C and N cycles to N fertilization in Sphagnum and Molinia-dominated peat mesocosms. Journal of Environmental Sciences (China) 77, 264–272. doi:10.1016/j.jes.2018.08.003

Lloyd, D., Thomas, K.L., Benstead, J., Davies, K.L., Lloyd, S.H., Arah, J.R.M., Stephen, K.D., 1998. Methanogenesis and CO2 exchange in an ombrotrophic peat bog. Atmospheric Environment 32, 3229–3238. doi:10.1016/S1352-2310(97)00481-0

---

## Author Response (AR2)

**Report #1**

pg3 line 6: revise the language: All mesocosms were choose because they was entirely covers by Sphagnum rubellum and without Molinia stems.
  ➢ It has been revised : "All mesocosms were entirely and exclusively covers by Sphagnum rubellum"

pg3 line7-8: you repeat the sentence twice: Molinia caerulea appeared in May and increased up to 60% of mesocosms on average until its senescence in November (Leroy et al., 2017) and did not affect Sphagnum cover (unpublished data).
  ➢ It has been corrected

pg7 line 12: you earlier use term PPFD and now PAR, I think it would be better to be consistent with this
  ➢ Yes, the term PAR have been replace by PPFD

pg12 line 4: " The parameters of the CH4 models differed with vegetation cover." if Iam not wrong (as Iam non-native English speaker), this implies that you had different set of parameters for the to vegetation types, but don't you try to say that the parameter values, or estimates differed?
  ➢ Indeed, parameters differed with vegetation covers and the modification of the temperature sensitivity are explained L5-6 (hypothesis). However, the others parameters had an opposite sign in the two vegetation covers and was difficult to interpret as the large variation of parameter e

page 12 l20-> I was not able to find the authors response to the previous round of reviewer comments so I come back with the same question. In introduction and what is referred to here also the expectation is that sphagnum is replaced by vascular plants, which could decrease the CO2 balance. In your study sphagnum was not replaced by Molinia, but Molinia was added into the system. so in theory, the question is bit different than what is introduced based on literature. In your case sphagnum functions likely stay unaltered, but Molinia adds up, so it is natural that co2 fluxes are higher. This is maybe a minor issue, but I think it would be good to repeat here that sphagnum cover was similar under both treatments.
  ➢ The sentence in introduction L5 have been slightly reformulated to make it more clear

**Report #2**

1. Explain the differences between this study and the two previous studies that use the same dataset. How and why are they different? Why does Table 1 in each paper have slightly different values for in-common parameters? Why was it not appropriate to include the current analyses in one or the other previous publication? This explanation should take up several paragraphs across the entire manuscript, including Introduction, Methods, and Discussion.
  ➢ Explanations are provided p2 L21 to 28; p3 L 19-21

2. Rigorous proof-reading and careful attention to English writing. Some of the mistakes would have been easily caught by any proof-reader, almost regardless of English skill level, such as the complete copy of a sentence.
  ➢ The manuscript has been corrected by a proof-reader

[revised manuscript text omitted]